

# Effects of disputes and easement violations on the cost-effectiveness of land conservation

Richard Schuster and Peter Arcese

Department of Forest and Conservation Sciences, University of British Columbia,
Vancouver British Columbia, Canada

## ABSTRACT

Conservation initiatives to protect and restore valued species communities in human-dominated landscapes face challenges linked to their potential costs. Conservation easements on private land may represent a cost-effective alternative to land purchase, but long-term costs to monitor and enforce easements, or defend legal challenges, remain uncertain. We explored the cost-effectiveness of conservation easements, defined here as the fraction of the high-biodiversity landscape potentially protected via investment in easements versus land purchase. We show that easement violation and dispute rates substantially affect the estimated long-term cost-effectiveness of an easement versus land purchase strategy. Our results suggest that conservation easements can outperform land purchase as a strategy to protect biodiversity as long as the rate of disputes and legal challenges is low, pointing to a critical need for monitoring data to reduce costs and maximize the value of conservation investments.

## INTRODUCTION

Despite an urgent need to develop mechanisms to promote biodiversity conservation (*Ehrlich & Pringle, 2008*; *Butchart et al., 2010*; *Bayon & Jenkins, 2010*; *Estes et al., 2011*), developing such mechanisms in human-dominated landscapes, where private ownership prevails and the cost of land purchase and opportunity costs of conservation can be substantial, is challenging (*Naidoo et al., 2006*; *Wunder, 2007*). One potentially cost-effective route to conservation in such areas may be to promote private land conservation easements or covenants that prohibit land use changes likely to reduce conservation values in exchange for monetary or other compensation (*Merenlender et al., 2004*; *Knight et al., 2011*). Advantages of easements include their low initial cost compared to land purchase (*Pence, Botha & Turpie, 2003*) and their ability to facilitate voluntary conservation with landowners wishing to retain title (*Langholz, & Lassoie, 2001*; *Winter, Esler & Kidd, 2005*; *Knight et al., 2010*; *Selinske et al., 2015*). Easements have thus gained global attention as conservation tools (*Fishburn et al., 2009*; *Gordon et al., 2011*). For the purposes of this study we define conservation easements, also called conservation

Corresponding author
Richard Schuster,
mail@richard-schuster.com

covenants in e.g., Canada and Australia, as voluntary agreements between easement holders, typically land trusts or government agencies, and private landowners (*Rissman et al., 2007*). Easement holders acquire and hold certain property rights in order to restrict land use, in most cases permanently (*Gustanski & Squires, 2000*; *Owley, 2004*). In return, private landowners may receive a payment and/or reduction in taxes (*Byers & Ponte, 2005*).

The factors affecting the cost-effectiveness of easement versus land purchase strategies for biodiversity conservation remain unclear (*Armsworth & Sanchirico, 2008*; *Fishburn et al., 2009*), but it is reasonable to assume that factors like the knowledge and willingness of landholders to manage their easements, as well as the types of ecosystems they are trying to manage would be part of this (*Selinske et al., 2015*). Because the long-term costs of monitoring, enforcing and legally defending easements is unknown, simple comparisons of land purchase to easements as approaches to conservation could overestimate the cost-effectiveness of easements (*Copeland et al., 2013*; *Morzaria-Luna et al., 2014*). Thus, representing these uncertainties as potential risks to easement holders should help estimate the cost-effectiveness of these alternate approaches to conservation (*Byers & Ponte, 2005*; *Knight et al., 2010*; *Rissman & Butsic, 2011*). The cost-effectiveness of easements is likely related to the extent of monitoring, which in itself is likely to be a substantial influence on the identification of easement issues, and the extent of recording those issues would influence how it becomes a violation or escalates to a dispute. Despite some effort to address financial risk, such as the 'Conservation Defense Liability Insurance' for easement holders in the United States (*Land Trust Alliance, 2011*), detailed studies are lacking and it remains unknown whether easements offer similar levels of biodiversity protection as compared to land purchase (*Merenlender et al., 2004*; *Fishburn et al., 2009*), despite increased demand for easement like agreements from landholders (*Fitzsimons & Carr, 2014*). These uncertainties highlight the need for a theoretical framework to evaluate the cost-effectiveness of biodiversity conservation by establishing conservation easements versus fee simple land purchase.

We developed a simple theoretical framework to explore conservation outcomes by defining and estimating the uncertainties described above, and using biodiversity and property values to compare the total cost and effectiveness of land purchase versus conservation easements as strategies to protect critically endangered Old Forest and Savannah habitats of the Georgia Basin of south-western British Columbia, where <20% of the landscape is owned by governments and only 9% allocated to conservation. Specifically, we asked two questions about the long-term (100 year) cost and biodiversity value of easement versus land purchase strategies: (1) how will dispute rate influence the cost-effectiveness of each approach, and (2) assuming that violations reduce the area of easements under conservation (e.g., *Smith, 2009*), what is the total area of the high-biodiversity landscape likely to be protected given alternative investment in easements versus land purchase. Here we define dispute as an attempt to or the active breach of the easement conditions by a landowner, such as illegal logging or attempting to add a new structure. As a consequence, this would lead to a reduction of habitat protected under the easement. We define a violation as a breach of easement conditions resulting in

a partial loss of the easement area, where the magnitude of the loss is dependent on the dispute cost. To answer these questions, we used occupancy maps for 47 bird species and expert elicitation to map high-biodiversity landscapes and assessed land value to represent cost. We then contrasted the land purchase scenarios of *Schuster, Martin & Arcese (2014)*, aimed at maximizing biodiversity conservation, to scenarios employing easements but subject to a range of assumptions about dispute rate and cost. *Schuster, Martin & Arcese (2014)* identified conservation networks based on fee-simple land purchase, designed to maximize avian biodiversity in Old Forest and Savannah habitats and found that using beta-diversity was the most cost-effective way to reach conservation targets. Overall, we offer a novel theoretical framework for evaluating land acquisition for biodiversity conservation and highlight the need for empirical analyses to estimate the long-term costs of monitoring and potential costs and rate of legal challenges.

## MATERIALS AND METHODS

### Ethics statement

Permits or permission for the use of bird point count locations were obtained from Parks Canada (locations in National Park Reserves), private land owners (locations on private land), or did not require specific permission as they occurred on public right of ways (e.g., roadsides, regional parks). As private land owners did not want their information posted publically, please contact the authors for contact details. The field studies did not involve endangered or protected species. This study did not require approval from an Animal Care and Use Committee because it was a non-invasive observational field study, and did not involve the capture and handling of wild animals.

### Study region

We studied a 2,520 km$^2$ portion of the Coastal Douglas Fir (CDF) ecological zone of the Georgia Basin of British Columbia (BC), Canada (Appendix S1). The CDF includes a critically endangered but diverse suite of old forest and savannah plant and animal communities endemic to the region but is ≥60% converted to human use (*Austin et al., 2008*), and ≤0.3% of old forest (>250 years) (*MES, 2008*) and ≤10% of oak woodlands extant prior to European contact remain (*Lea, 2006*), both of which provide habitat for 117 species at risk of extirpation (*Austin et al., 2008*). Because regional, provincial and federal authorities own <20% of the region and only ~9% is already conserved, cost-efficient routes to conservation are urgently needed.

Prior to European colonization the CDF occurred as uneven-aged forest (often >300 years old) dissected by shallow and deep-soil meadow and woodland communities (*Meidinger & Pojar, 1991*; *Mosseler, Thompson & Pendrel, 2003*) maintained in part by aboriginal land management practices to enhance hunting opportunities and root and fruit harvests (*MacDougall, Beckwith & Maslovat, 2004*; *Dunwiddie & Bakker, 2011*; *McCune, Pellatt & Velend, 2013*; *Turner, 2014*). Oak woodland and savannah community distributions are predicted to shift under future climate conditions, and only a small fraction of the current protected areas have the potential to accommodate this shift
(*Pellatt et al., 2012*). There is large land use heterogeneity within the region and the potential for humans to directly or indirectly affect native species richness (*Gonzales & Arcese, 2008*; *Martin, Arcese & Scheerder, 2011*; *Bennett et al., 2012*; *Schuster & Arcese, 2013*). Several land trusts in the region, such as The Nature Trust of British Columbia (http://www.naturetrust.bc.ca) and the Islands Trust Fund (http://www.islandstrustfund.bc.ca) hold conservation easement. Protected areas in the region range from 0.01 to 5,830 acres (mean = 81) in size and span whole islands of up to 1,300 acres in size, over small parcels in urban areas to holdings in remote (island) locations. In general, real estate in the region is very expensive with an average cost per acre of $2.1M, but there are affordable properties in non-urban areas (8% of all properties cost less than $100,000 per acre).

## Land purchase cost scenario

We built on *Schuster, Martin & Arcese (2014)* to identify conservation networks based on fee-simple land purchase and designed to maximize avian biodiversity in Old Forest and Savannah habitats. To do so, we developed distribution models for 47 birds based on 25 remote-sensed predictor variables and incorporating imperfect detectability (*Mackenzie et al., 2002*) to create composite community scores (*Schuster & Arcese, 2013*). To consolidate focal species occurrence predictions into an index of Old Forest and Savannah community association, we combined focal species occurrence predictions using expert elicitation to rank the likelihood of observing 47 species in 10 focal habitat types using photographic and text descriptions of herbaceous, shrub, woodland, wetland, four forest types (pole, young, mature and old), and 2 human-dominated habitats (rural, urban), to create two community metrics indicating Old Forest and Savannah (SAV) habitats (*Schuster, Martin & Arcese, 2014*). These metrics match our goals given the region's history and focus on Old Forest and Savannah community conservation (see 'Study region'). Specifically, each species contributed to the cumulative Old Forest or Savannah community score, weighted by its expert opinion score for the given sub-type, summed across species to create community specific association scores from 0 to 1, and corresponding to none versus all members of the community expected to be present. We then combined Old Forest and Savannah community scores to create a beta-diversity metric to identify heterogeneous landscapes likely to maximize the occurrence of both target communities, as previous work has shown that using a beta-diversity metric results in the most cost-effective reserve design solutions (*Schuster, Martin & Arcese, 2014*).

Cadastral data was used to identify properties and 2012 assessments to represent property cost. We then used the systematic reserve design software Marxan (*Ball, Possingham & Watts, 2009*) to prioritize properties ($n = 193,623$) by the beta-diversity metric for inclusion in conservation networks to protect 20% of the total beta-diversity scores (*Schuster, Martin & Arcese, 2014*). We retained 100 Marxan solutions to estimate variability in spatial network configuration and cost. As acquired properties require investments in staff, infrastructure and land management over time, it has been suggested that land trust use endowments to fund these costs in the long term (*Armsworth et al., 2015*). Most large land management organizations in the region have recently adopted a policy of raising

funds for each land purchase that equal 15% of purchase value, and put those funds in trust (The Nature Trust of British Columbia, pers. comm., 2015). To account for this, we added 15% to purchase prices for each property in a conservation network. Disputes that effectively increase land purchase costs are reported from parts of the US (*Rissman & Butsic, 2011*), but so far none have been reported in our study region; thus we excluded dispute costs from our land purchase scenario. This decision was based on an informal survey conducted at The Nature Trust of British Columbia, which serves as the curator of the Land Trust conservation land database, which yielded no reported disputes. Another potential issue for the long term conservation of biodiversity by protected areas is downgrading, downsizing and degazettment (PADDD) (*Mascia & Pailler, 2011*). However, in the study region no cases of PADDD have been reported yet (Islands Trust Fund, pers. comm., 2015) and the Best Practices for Land Trusts in Canada requires Land Trusts to monitor protected lands and manage damages (*Canadian Land Trust Alliance, 2005*), which can be dealt with via the endowed management funds mentioned above if they do occur.

## Easement cost metrics and assumptions

All properties selected in land purchase Marxan solutions were used as candidate easements under the assumption of willing current owners for both land purchase and easement participation. We did not estimate change in land value given easements as there is no consensus on magnitude or direction (*Anderson & Weinhold, 2008*). Easement costs used here reflect the experience of The Nature Trust of British Columbia and Islands Trust Fund and published examples (*Main, Roka & Noss, 1999*; *Parker, 2004*). We compiled estimates of fixed easement costs including: legal, financial advice, registration and endowment fees, as well as scalable costs of surveys and appraisal (Table 1). Land managers identified reoccurring costs of annual monitoring and staff time to address land owner requests (Table 1). All costs were estimated in present day Canadian dollars, because the alternative use of a discount rate equal to the inflation rate for costs incurred over time, and then reporting in future dollar values is highly sensitive to the discount rate, adding substantial uncertainty on future dollar amounts (*Arrow et al., 2013*).

## Conservation easement scenarios

We calculated the cost-effectiveness of alternate scenarios as the fraction of the high-biodiversity landscape protected, divided by the total reserve network cost for each scenario (*Wilson et al., 2007*) and then standardized this value by the cost of land purchase for comparisons. We followed *Rissman & Butsic (2011)* to estimate the distribution of dispute costs and fitted a cost profile bound between $1,000 and $400,000 following the power function $cost[\$] = 4{,}845.78 * disputes^{-0.701}$. We also explored cost profiles including dispute costs over $400k using a truncated normal distribution where the left tail was cut off. As the starting point or mean of that distribution, we used $400k. We chose a very wide standard error ($1M) to allow for dispute costs to rise substantially above $400k, and 1% probability of those costs arising but found similar results, and thus restricted our analysis to published values. To find the easement dispute rate that caused the cost effectiveness of land purchase to exceed that of easements, we used dispute rates of 0.028,

**Table 1 Easement cost estimates from The Nature Trust of British Columbia and Islands Trust Fund.** All variable costs follow a saturating curve in the form of: cost = Intercept + Slope * ln(easement size [acres]), with the constraint that the cost cannot fall below the minimum.

|  | Cost [$] |
|---|---|
| **Fixed costs** | |
| *Land owner* | |
| Legal cost | 300 |
| Financial advice | 300 |
| Easement registration | 200 |
| Endowment | 10,000 |
| *Easement holder* | |
| Legal cost | 4,000 |
| **Variable costs (by easement size)** | |
| *Ecological baseline (minimum $1,000)* | Cost = 2,185+ 1,957 * log(easement size in acres) |
| *Appraisal (minimum $1,500)* | Cost = 0 + 1,957 * log(easement size in acres) |
| *Land survey (minimum $1,000)* | Cost = 300 + 1,957 * log(easement size in acres) |
| **Reoccurring costs (yearly)** | |
| Easement monitoring | 758 |
| Staff cost to reply to Land owner request | 151.6 |

0.28 and 2.8% of easements per year. *Rissman & Butsic (2011)* surveyed 205 land trusts to report that they experienced about 2.8 disputes per year, but because they did not record the total number of easements represented we could not estimate dispute rates precisely.

Typically, if an easement is held by an organization, and a landowner breaks the easement in a way that reduces habitat, such as via illegal logging, the easement holder must sue for damages. There was no data on disputes available for the study area, but a 1999 Land Trust Alliance survey found that out of 7,400 conservation easements 498 violations were reported, 383 of which were minor and got resolved without significant commitment of resources. 115 were major violations, but 94 were resolved without litigation and 21 cases resulted in lawsuits (*Danskin, 2000*). Suits can be costly but if the organization wins, which happened in all but one case reported by *Danskin (2000)*, one of two things may happen: 1. landowners pay restitution to the organization, or 2. are forced to restore the damage. We therefore included two types of disputes in our models: pre-emptive enforcement (PE), and damage enforcement (DE). PE occurs when a landowner attempts to do something disallowed by the easement, such as adding as new structure, and the easement holder sues to stop them. The outcome here may be little or no habitat change but a large cost to the easement holder to enforce the easement. DE involves activities that have already taken place, such as illegal logging. In this case, easement holders may sue successfully for compensation of their loss, thus with no monetary cost to the dispute, but still suffer habitat damage and biodiversity decline.

In each year of our simulation, easements suffered disputes at rates assumed above and, given a dispute, were randomly assigned to either the PE or DE dispute model. In the PE case, an easement was assigned a randomly drawn dispute cost that contributed to the total

**Table 2 Main results summary.** Summary of the main results related to cost and biodiversity loss. Presented are the mean values with min–max range in brackets. Biodiversity loss values are calculated from the initially protected biodiversity values, i.e., those values represent 100%.

| | Cost (Million $) | Biodiversity loss (%) |
|---|---|---|
| Baseline: Land purchase cost (15% in trust added) | 457 (441–470) | 0 |
| Easement setup cost | 44 (43–45) | 0 |
| Easement cost after 100 years with no disputes | 162 (157–166) | 0 |
| Low dispute rate (0.028% of easements/year) | 162 (157–167) | 0.4 (0.1–0.8) |
| Medium dispute rate (0.28% of easements/year) | 180 (173–189) | 3.7 (2.2–5.2) |
| High dispute rate (2.8% of easement/year) | 355 (339–382) | 31.7 (28.9–36.1) |

cost of easement scenarios. To quantify the effect of disputes on biodiversity values in a DE case we assumed that biodiversity loss followed the same distribution as dispute cost, bounded between 0 and 100%, which was then used to reduce the disputed easement's beta diversity metric. In the absence of empirical study, we also relaxed that assumption by allowing variation in biodiversity loss to follow a normal distribution around the estimate (SD = 5% of total biodiversity loss possible). All analyses were conducted using R v.3.2.0 (*R Core Team, 2015*) and the analysis script can be found in Appendix S1. We have also provided a reproducible example of the entire analysis on GitHub (https://github.com/yeronimo/Easement_cost-effectiveness or http://dx.doi.org/10.5281/zenodo.21144).

## RESULTS

Given a goal of protecting 20% of the high-biodiversity landscape, land purchase scenarios protected a mean of 370 km$^2$ (range = 365–374 km$^2$) at a mean cost of $457M (range = $441–470M) (Fig. 1A). In comparison, the cost of an equivalent area under conservation easements averaged $43.9M in year 1 (range = $42.6–45.0M) and $162M cumulatively to year 100 (range = $157–166M; Fig. 1A), representing a 65% reduction in cost compared to land purchase. Including dispute rates of 0.028 and 0.28% increased long term costs in easement scenarios by 0.5 and 11%, respectively (Fig. 1A). However, with 2.8% of easements experiencing disputes annually, network cost increased up to 219% (mean = $355M, range = $339–382M), but this was still below the cost of land purchase (Fig. 1A). Table 2 summarizes our overall results.

Baseline scenarios in the absence of disputes aimed to protect 20% of the high-biodiversity landscape. However, under the assumption that disputes cause biodiversity loss, a dispute rate of 0.028% reduced the area effectively conserved after 100 years by 0.36% (range = 0.05–0.83%) compared to baseline (Fig. 1B). In contrast, an intermediate dispute rate (0.28%) returned a mean reduction of 3.71% (range = 2.19–5.18%), and a high dispute rate (2.8%) returned a mean reduction of 31.73% (range = 28.94–36.1%; Fig. 1B).

Given our results above, the cost-effectiveness of conservation easement versus land purchase scenarios was 2.8–3.3 times higher after 100 years (Fig. 2). However, a high annual dispute rate of 2.8% caused the cost-effectiveness of easement scenarios to approach that of land purchase (Fig. 2).

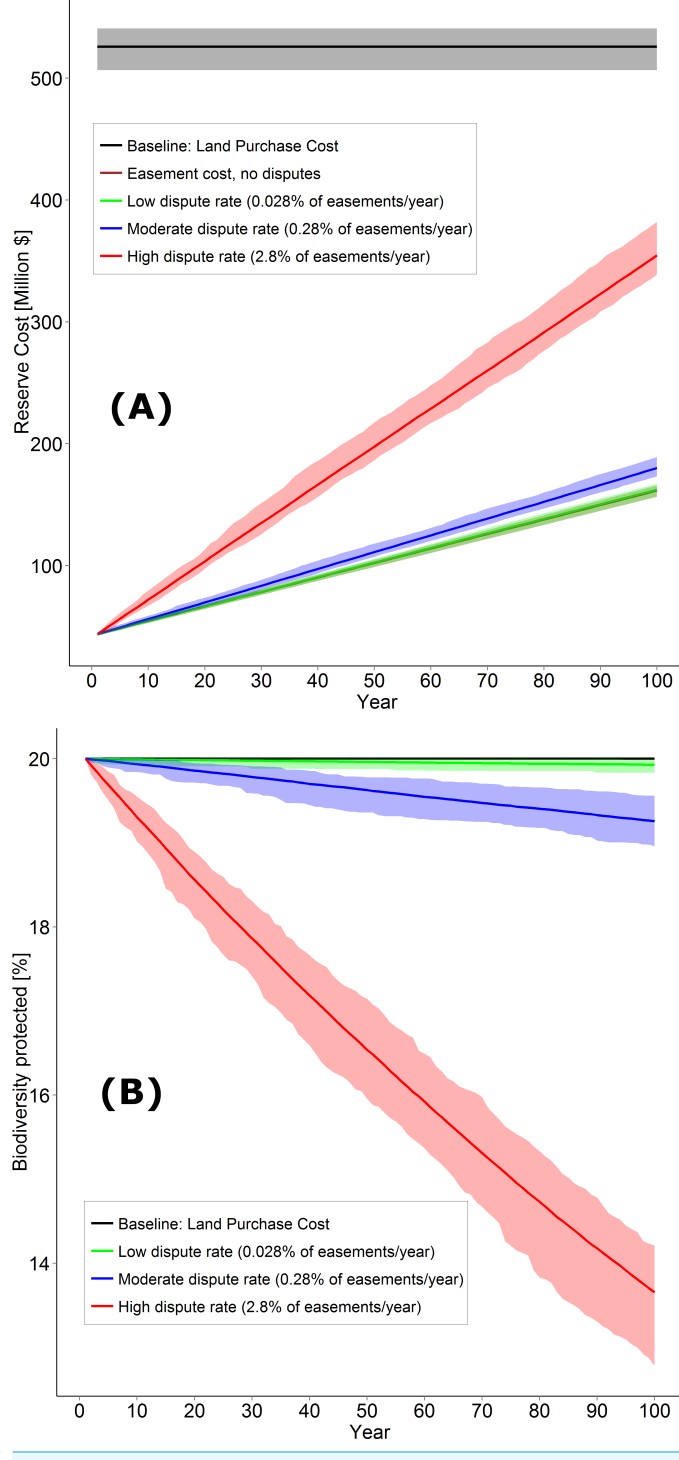

**Figure 1 Cost and biodiversity loss comparisons.** (A) Conservation network cost comparison between land acquisition and conservation easements of varying dispute rates. (B) Biodiversity loss of varying easement dispute rates in conservation networks and an initial 20% protection level of current biodiversity in the CDF ecological zone. Solid lines represent mean values for each approach, and the corresponding ribbons show minimum and maximum values for the 100 Marxan solutions.

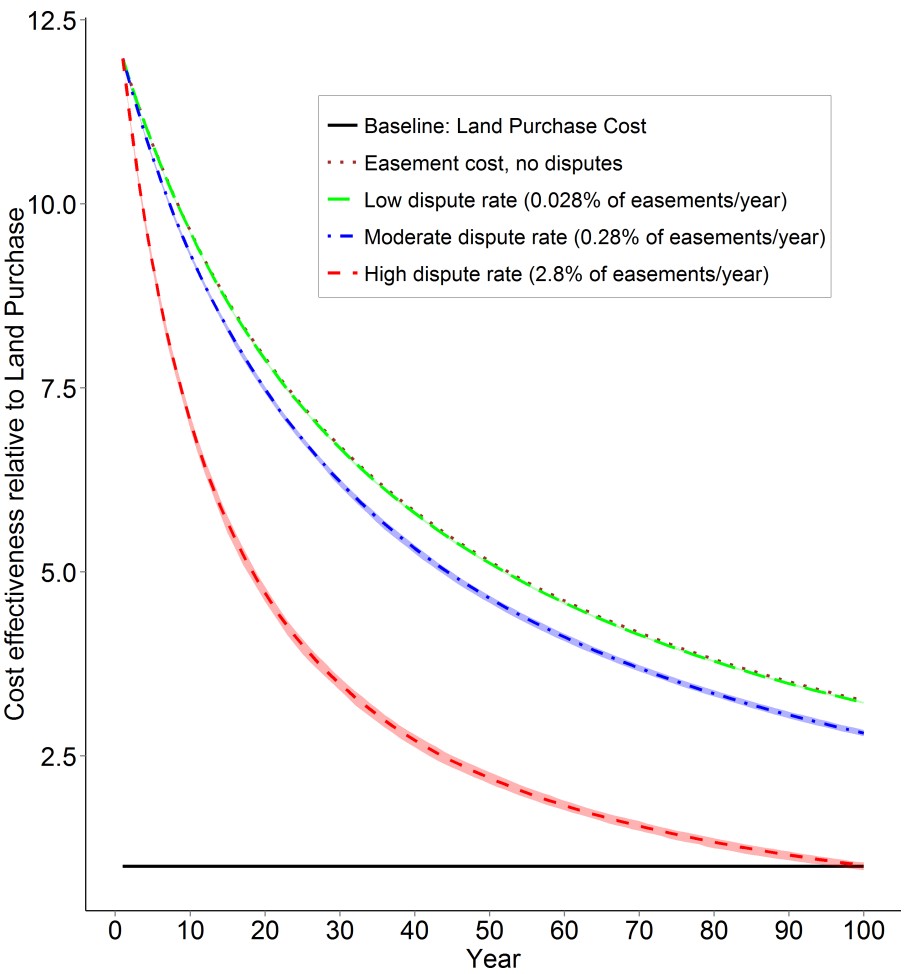

**Figure 2 Conservation Easement cost effectiveness.** Long term cost effectiveness defined as rate of biodiversity protected divided by the reserve network cost. Values are relative to the baseline land purchase scenario. Solid lines represent mean values for each scenario, and the corresponding ribbons show minimum and maximum values for the 100 Marxan solutions.

## DISCUSSION

We show that easement violations and disputes can substantially affect the long-term cost-effectiveness of conservation strategies employing easements or land purchase to protect high-biodiversity landscapes. In particular, in terms of cost-effectiveness land purchase performed as well as an easement approach to protection when dispute rates were high, in part because disputes often co-occur with biodiversity loss (Figs. 1B and 2). These results point out critical uncertainties about the cost-effectiveness of conservation easements and potential liabilities to easement holders. However, the low initial cost of conservation easements as compared to land purchase suggest that as long as disputes are rare, easements should outperform land purchase as a cost-effective tool for biodiversity conservation (Fig. 2). We now develop these points in light of literature on land acquisition and easements and identify several remaining uncertainties.

## Easement dispute rate

We found that the cost-effectiveness of easements versus land purchase depend primarily on easement dispute rate (Fig. 2), indicating that minimizing dispute rate should be a key goal. However, the paucity of published data on the frequency and cost of disputes (*Byers & Ponte, 2005*; *Rissman & Butsic, 2011*) points to an urgent need to formalize the experience of conservation organizations to identify pitfalls and reduce dispute rate in future. A survey conducted by the Land Trust Alliance showed that dispute rate increases with the number of successive owners of easement properties (*Jay, 2014*). If true, some existing easements may represent unrecognized risk to holders that should be remedied before titles transfer.

## Dispute costs

We adopted a dispute cost profile based on a survey of 205 land trusts, but including substantial uncertainty and a maximum dispute cost of $400k (*Rissman & Butsic, 2011*), but are aware of examples with potentially much higher costs. Although we used an inverse dispute cost profile in our simulations, the risk of very large costs remains an uncertain risk to all easement holders. Empirical data are therefore needed to characterize cost profiles and facilitate realistic analyses (*Boyd, Caballero & Simpson, 1999*; *Game et al., 2013*). Although more complex cost profiles than we used can be imagined, they remain speculative in the absence of data and we observed modest variation in dispute costs had little influence on our results. We used pre-emptive enforcement (PE) and damage enforcement (DE) to represent possible legal scenarios here. PE occurs when a landowner attempts to do something disallowed by the easement, such as adding as new structure, and the easement holder sues to stop them. The outcome here may be little or no habitat change but a large cost to the easement holder to enforce the easement. DE involves activities that have already taken place, such as illegal logging. In this case, easement holders may sue successfully for compensation of their loss, thus with no monetary cost to the dispute, but still suffer habitat damage and biodiversity decline. We believe PE and DE represent reasonable approximations of the most common dispute scenarios, but acknowledge different scenarios such as DE without compensation would be possible as well, but less likely.

## Biodiversity loss and easement dispute

The potential for biodiversity loss when easements are violated is also uncertain. We found that at intermediate dispute rates, the area of high-biodiversity landscape conserved declined 3% after 100 years (Fig. 2). However, at higher dispute rates nearly a third of the conserved landscape was lost, assuming disputes involve the loss of protected elements or site integrity (*Smith, 2009*). Although our assumption that biodiversity loss and dispute cost vary directly remains untested, we suggest it is a reasonable initial assumption given that easement violations often involve land clearing, road building or new structures likely to reduce biodiversity value (*Danskin, 2000*). Thus, easement disputes have the potential to add management cost and facilitate biodiversity loss, suggesting these potential costs must be considered when comparing conservation strategies.

## CONCLUSION

We offer a theoretical approach to compare the cost-effectiveness of conservation easements vs. land purchase as alternative approaches to biodiversity conservation. Our results indicate that over the long-term, the cost-effectiveness of conservation easements should outperform land purchase as a strategy to protect habitat as long as the rate of disputes and legal challenges to easements remain low. Anecdotal evidence from land trusts suggests that the risk of disputes depends a lot on how the covenants were written, and they have and are continuing to get better. We think that improved legal wording will reduce risk of easement failure over the long-term and could help conservation organizations to save guard against legal disputes. Recommended actions to improve the comparison of land purchase and easements as approaches to biodiversity protection include estimating more precisely (i) easement dispute rates and cost profiles over time, and (ii) biodiversity loss given a dispute. Our findings should apply generally to landscapes with high rates of private ownership and human impact, such as the Georgia Basin of western North America.

## ACKNOWLEDGEMENTS

We thank the Conservation Decisions Team (CSIRO, Australia) for hosting RS, S Gergel, R Germain and V LeMay and four anonymous reviewers for comments on an earlier version of this manuscript, The Nature Trust of British Columbia, Islands Trust and A Rissman for input on the easement process and costs.

### Funding

The Natural Sciences and Engineering Research Council, Canada, W and H Hesse, the University of British Columbia, and an Endeavour Research Fellowship, Australia (RS) funded this work. The funders had no role in study design, data collection and analysis, decision to publish, or preparation of the manuscript.

### Grant Disclosures

The following grant information was disclosed by the authors:
Natural Sciences and Engineering Research Council.
University of British Columbia.
Endeavour Research Fellowship.

### Competing Interests

The authors declare there are no competing interests.

### Author Contributions

- Richard Schuster conceived and designed the experiments, performed the experiments, analyzed the data, contributed reagents/materials/analysis tools, wrote the paper, prepared figures and/or tables, reviewed drafts of the paper.

● Peter Arcese conceived and designed the experiments, performed the experiments, contributed reagents/materials/analysis tools, wrote the paper, reviewed drafts of the paper.

## Animal Ethics

The following information was supplied relating to ethical approvals (i.e., approving body and any reference numbers):

Permits or permission for the use of bird point count locations were obtained from Parks Canada (locations in National Park Reserves), private land owners (locations on private land), or did not require specific permission as they occurred on public right of ways (e.g., roadsides, regional parks). As private land owners did not want their information posted publically, please contact the authors for contact details. The field studies did not involve endangered or protected species. This study did not require approval from an Animal Care and Use Committee because it was a non-invasive observational field study, and did not involve the capture and handling of wild animals.

## Data Availability

The following information was supplied regarding the deposition of related data:

https://github.com/yeronimo/Easement_cost-effectiveness/
DOI: 10.5281/zenodo.21144.

## Supplemental Information

Supplemental information for this article can be found online at http://dx.doi.org/10.7717/peerj.1185#supplemental-information.

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
