# Peer review of "Effects of disputes and easement violations on the cost-effectiveness of land conservation"

_PeerJ, doi:10.7717/peerj.1185_

## Round 0.1 · original submission · Minor Revisions

· Academic Editor

Minor Revisions

As you can see the reviewers are generally positive about your manuscript, however reviewer 1 in particular has identified a number of areas that need clarification.

Reviewer 1 ·

Basic reporting

The article does not include sufficient introduction and background to demonstrate how the work fits into the broader field of knowledge. The submission is relatively self-contained, but the methodology leans heavily on a previous study by the authors that needs to be summarised in this paper.

In general the following areas need to be improved:

Introduction
• Introducing and clarifying easement disputes and violations, esp. their differences and why they’re important for consideration in assessing cost-effectiveness
• Justifying the comparison between land acquisition and easements
• Setting this paper up as part of a bigger question over land acquisition vs easements

Methods
• Including more information on the case study, including some context on easements in Canada

Results
• Using separate line types for each curve to assist readers viewing greyscale / black and white versions of the manuscript

Discussion
• Enhancing the interpretation of the results to a wider audience, and providing greater contextualisation of the study’s findings using existing literature
• Ensuring that the findings discussed here are consistent with the results presented


Specifically:
Line 1 - Title – easement failure could be understood in a number of ways, ecological failure, legal failure, financial failure. I suggest refining the title to more closely reflect the scope of the article – that of the effect of disputes and easement violations on cost-effectivenessLines 30-31 – what makes it challenging?
Lines 31-38 – Although the detail of US easements are relatively well known I would like to see more detail on easements in Canada, including how they’ve been used. Are they different to those in the US and elsewhere?
Lines 47-48 – what efforts have been taken to address financial risk, and to whom? Easement holders? The landholders?
Lines 50-51 – Fitzsimons and Carr (2014) talk about demand for covenants by landholders – which are very similar to easements but slightly different. I suggest to use a more nuanced term here along the lines of –“easement-like agreements” and just clarify that the demand stated in this article is from landholders.
Lines 40-53 – I imagine that there a whole range of factors that influence the cost-effectiveness of easements – including the knowledge and willingness of landholders to manage their easements, as well as the types of ecosystems they are trying to manage.
Lines 40-53 – Presumably the cost-effectiveness of easements is related to the extent of monitoring, which in itself is likely to be a substantial influence on the identification of easement issues, and the extent of recording those issues would influence how it becomes a violation or escalates to a dispute.
Lines 61-62 - I would recommend defining what is meant by “dispute”– is this about landowners challenging their easements in court? Or simply refusing to adhere to their obligations and/or actively breaching the easement conditions? What contributes to disputes? And what is the end-point of a dispute? This would help the reader understand what a dispute might look like, and what elements your evaluation of cost-effectiveness might need to include (e.g. court costs, staff time, monitoring)
Lines 62-64 – what is meant by violations (presumably that the easement conditions have been breached)? If so, how does this relate to positive or negative easement obligations? And what is the end-point of a violation, a total loss of the easement area, or partial?
Lines 62-64 – Is there a study you can reference that shows violations leading to a reduction in area under conservation? I imagine that in some cases this is true, but in many cases easement holders may wish to deal with easement violations in other ways (as you talk about later in the paper).
Line 66-68 – The study relies quite heavily on comparison with the land purchase scenarios in Schuster et al. (2014). The manuscript would be improved by including a summary of these study scenarios - what are they? What was found?
Lines 73-91 – I would like to see more detail to help understand the case study – what is the study area like? What are the competing land uses? Are easements currently used in this area, and are they unique in any way? Are their specific management challenges for easement holders? Is real estate expensive? What do land acquisitions typically look like – are they large holdings or small? Expensive properties or in marginal areas that don’t attract much real estate demand?
Lines 196-197 – you state that land purchase performed as well as easements when dispute rates were high. Doesn’t this contradict your results that state even with the high dispute rate of 2.8%, the annual easement network cost was below the cost of land purchase (Lines 176-178)?
Lines 200-202 – again, this seems to contradict what you stated in your results – even with disputes, easements seem to outperform land purchase?
Lines 202-203 – for the discussion in general, and perhaps the conclusion, from a practitioner point of view is there anything that easement holders can do to keep dispute costs down?
Line 210-211 – there are studies in the literature that could help back up this statement beyond anecdotal evidence – e.g. Collins 2000; Gustanski 2000:16
Line 216-223 – although the dispute costs could be high, presumably there would/could be cases where disputes could be avoided (see Selinske et al 2014) or dispute costs could be recouped – i.e. where the easement holder wins the case.
Lines 231-232 – examples of easement violations - are there any studies that you could reference here?
Lines 241-245 – are there recommendations for easement holders to increase the cost-effectiveness of their easements (i.e. keep dispute rates down?)

Experimental design

The methods are not present sufficient to be reproducible by another investigator.

In general, the methods could be improved by:
• Providing further detail on the methodology followed to increase reproducibility
• Providing a short summary of findings from Schuster et al 2014 model, which seems to form an important part of the methodology
• Providing greater clarity on the scenarios investigated – perhaps a table or specific headings would help here?

Specifically:

Lines 107-108 – admittedly I don’t know much about the costs associated with land acquisition management - but 15% of purchase cost doesn’t sound like enough to cover ongoing management over the timescale of this study – is there anything you could reference here? And are there any other sources of funds that contribute to ongoing management that should be considered?
Line 110-114 – as this sets up your baseline, this assumption has the potential to dramatically change your results, and also raises questions about their comparability if acquisition doesn’t include dispute costs – is there any detail you can provide on how you got this information and what negligible means?
Line 118 – please clarify which cases you are referring to.
Line 137 – are you stating a mean of $400k and a standard error of $1 million? Isn’t the standard error too large?
Lines 145-147 – Do we have a sense of how many of these violations do easement holders end up pursuing? i.e. presumably most violations are likely to be small, with fewer being large-scale and thus worthy of taking the landholder to court?
Lines 147-148 – there is an assumption here that the easement holder will win, but I am wondering if it’s important to include some justification for this – for example, is there any information on how many cases there are, and how many are actually won by organisations?

Validity of the findings

No comments

Additional comments

This manuscript outlines the long-term cost effectiveness of conservation easements in comparison with land acquisition, based on a case study of south-western British Columbia, Canada. The topic of this paper is of importance and will be of interest to practitioners and policy-makers in private land conservation – quantifying the relative costs of conservation easements compared to that of land acquisition. This paper is one part of this bigger question. This paper has potential to contribute to the ongoing literary debate over the long-term effectiveness of private land conservation, and provide valuable insights for the development of private land conservation policy more specifically. However, the manuscript, whilst the methodology appears sound, in its current form is unpolished and in some areas lacking in detail, readability and structure. In general I was left feeling that the paper needs substantive revision, particularly clarifying the context, theoretical framework, scope (i.e. disputes), and the findings in light of related literature.

Reviewer 2 ·

Basic reporting

The manuscript is well written and clearly laid out. Two minor points:

1) I would suggest citing the new article by Selinske et al.

Selinske, M. J., Coetzee, J., Purnell, K., & Knight, A. T. (2015). Understanding the Motivations, Satisfaction, and Retention of Landowners in Private Land Conservation Programs. Conservation Letters, in press.

2) Figure 1b. Change "Biodiversity protected" to "Land under conservation"

Experimental design

The experimental design seems fine.

The only thing to flag up is that I don't think the Marxan analysis is the best way of using the data but I don't think this is a fatal flaw, especially as their approach has already been published in PLOS ONE and just provided an input layer for this analyses.

My concern with the Marxan analysis is that, from my reading of the methods, the only conservation feature data they used was a beta diversity metric. This risks choosing areas that are very similar in terms of species composition that fail to conserve the species themselves. Given the authors had data on the distribution of each species, I think it would have been much better to set targets for each species as well as beta diversity (and ideally have "high beta diversity habitat" as a conservation feature, rather than the raw numbers).

But as I said, I don't think this is a fatal flaw for this paper. And if I have misinterpreted the analysis then the authors should rewrite the description to make things clearer.

Validity of the findings

The findings seem valid.

One issue that I wondered about was the extent to which land that is purchased for conservation is also damaged, either by external actors or because the state de-gazettes land. It would be interesting to ask the conservation agencies about whether this has happened in British Columbia and to discuss this as a potential problem (in the light of the literature on PADDD). If it is a potential problem, then it should be explained that the results illustrated in Fig 1b are overly positive about how well purchased land conserves biodiversity.

Additional comments

This is a fairly basic analysis with intuitive results but the manuscript highlights an important issue and acts as a good foundation for further research.

---

## Round 0.2 · accepted · Accept

· Academic Editor

Accept

Thanks for making the changes and the rebuttal letter was particularly clear and helpful.